**communications** engineering

# 4H silicon carbide bulk acoustic wave gyroscope with ultra-high *Q*-factor for on-chip inertial navigation
Zhenming Liu [1,2] ✉, Yaoyao Long[1], Charlotte Wehner[1], Haoran Wen[2] & Farrokh Ayazi[1,2] ✉

Inertial navigation on a chip has long been constrained by the noise and stability issues of micromechanical Coriolis gyroscopes, as silicon, the dominant material for microelectromechanical system devices, has reached the physical limits of its material properties. To address these challenges, this study explores silicon carbide, specifically its monocrystalline 4H polytype, as a substrate to improve gyroscope performance due to its low phonon Akhiezer dissipation and its isotropic hexagonal crystal lattice. We report on low-noise electrostatic acoustic resonant gyroscopes with mechanical quality factors exceeding several millions, fabricated on bonded 4H silicon carbide-on-insulator wafers. These gyroscopes operate using megahertz frequency bulk acoustic wave modes for large open-loop bandwidth and are tuned electrostatically using capacitive transducers created by wafer-level deep reactive ion etching. Experimental results show these gyroscopes achieve superior performance under various conditions and demonstrate higher quality factors at increased temperatures, enabling enhanced performance in an ovenized or high-temperature stabilized configuration.

In recent years, there has been increasing interest in high-performance inertial measurement units (IMU) due to the rapid growth of the market for autonomous vehicles, indoor navigation, and the metaverse[1–3]. The performance of a precision gyroscope, a crucial component in IMUs, can be categorized into rate, tactical, and navigation grades based on factors such as its full-scale range, operational bandwidth, angle random walk (ARW), and bias instability (BI)[4]. Currently, most tactical and navigation-grade gyroscopes depend on expensive and bulky optical devices[5,6], unsuitable for on-chip integration and with limited use cases. On the other hand, planar microelectromechanical system (MEMS) Coriolis gyroscopes with millimeter scale footprints offer advantages in terms of size, weight, power consumption, and cost (SWaPC), and can be integrated with almost any electronic devices, including GPS. However, their performance, especially the noise-related factors such as ARW and BI, falls short of their optical counterparts[7–10].

Using a high *Q* bulk acoustic wave (BAW) gyroscope has proven to be a good solution for achieving higher performance in MEMS gyroscopes[11–14]. When the frequencies of two gyroscopic modes of a BAW resonator overlap within their −3 dB bandwidth, the scale factor or sensitivity of the device is amplified by the mechanical quality factor, *Q*. Consequently, improving the *Q* factor in BAW resonators has been a focal point of research to advance MEMS gyroscopes. Currently, capacitive BAW resonators in silicon have reached nearly their physical limit set by small intrinsic phonon loss known as Akhiezer dissipation[14,15].

To achieve a breakthrough in the next-generation MEMS gyroscope, the selection of a substrate with a small Akhiezer dissipation is crucial. In prior work[16], we explore the potential of silicon carbide (SiC), particularly monocrystalline 4H-SiC, and present the design and implementation of capacitive SiC BAW disk resonant gyroscopes. This paper shall elaborate on the silicon carbide gyroscope operates at 3 MHz with an ovenized *Q* value of 4.6 million at 80 °C. The 3.5 μm capacitive gap was defined through wafer-level deep reactive ion etching (DRIE) on a bonded SiC-on-insulator (SiCOI) substrate. The resonant BAW gyroscope was mode-matched through electrostatic tuning and exhibited promising ARW of 0.005°·h$^{-1/2}$, BI of 0.34°·h$^{-1}$ in open-loop configuration, falling between the tactical and navigation grade performance level. Additionally, statistical data on as-born frequency response and *Q*, including intriguing temperature behavior obtained from the batch-processed devices, will be presented to shed light on the use of SiCOI platform for precision MEMS sensors.

## Results and discussions
### 4H-SiC BAW gyroscope
While SiC MEMS has drawn much attention over the past decades, most research was done on thin-film type devices, with performance and repeatability often limited by film quality and thickness. For example[17], has shown the Q factor of a 4H-SiC cantilever beam is 10X higher than that of a 3C-SiC cantilever beam, by improving the film growth method to achieve

[1]School of Electrical and Computer Engineering, Georgia Institute of Technology, Atlanta, GA 30308, USA. [2]StethX Microsystems Inc., Atlanta, GA 30308, USA. ✉e-mail: zhenming_liu@gatech.edu; farrokh.ayazi@ece.gatech.edu

fewer stacking faults and dislocations. In this work, we use a SiC-on-Insulator (SiCOI) process to enable fabrication of thick monocrystalline 4H-SiC devices in a similar way to conventional silicon-on-insulator (SOI) MEMS devices. The thick 4H-SiC layer results in a larger mass and a higher transduction efficiency compared to thin film devices, with low defects and stress.

Figure 1a, b demonstrate the schematic and cross-section of the 4H-SiC disk gyroscope fabricated on the SiCOI platform. 0.5-μm thick poly-crystalline silicon (Poly-Si) contacting pads were patterned on the electrode for ohmic contact formation to the 40-μm thick wide-bandgap SiC device layer. The resonator is centrally anchored with backside release holes etched into the silicon handle layer to release the structure. The demonstrated 4H-SiC disk resonators are operated in a pair of m = 3 BAW modes, as shown in Fig. 1c, owing to their mode degeneracy, superior decoupling from the central anchor and high Q. The electrode layout is designed based on the displacement of its mode shape: A total of 24 electrodes are uniformly placed around the disk resonator. The drive mode is actuated and read using $V_{drv}$ and $I_{drv}$ electrodes, placed at two antinodes opposite to each other; the differential readout $I_{sns+/-}$ of the sense mode is similarly placed at opposite antinodes and is 90° apart from the drive mode electrode; 4 pairs of frequency tuning electrode, $T_{1/2}$, are located at the remaining antinodes of corresponding mode; and 6 pairs of quadrature nulling electrode, $Q_{a/b}$ are between antinodes and nodes. The disk is centrally anchored with a folded beam decoupling network to isolate the resonator from the handle layer, eliminating the frequency and Q sensitivity to the mounting condition or thermal stress in the handle layer during ovenization (Fig. 1d). Figure 1e shows an optical picture of the device wirebonded onto a printed circuit board with the interface electronics schematic shown in Fig. 1f. The resonator input electrode is connected to the AC power output after a unit gain buffer. A transimpedance amplifier (TIA) with 100 kΩ gain amplifies the current from each output electrode and an instrumentation amplifier provides the differential response of the sense mode; a lock-in amplifier is used to build a phase-locked loop and synchronous demodulation architecture for the gyroscope control and readout.

While a similar resonator design has proven to be effective when implemented in (100) silicon[18], the design is more stable and robust in the 4H-SiC substrate owing to the different crystal lattice structure (Fig. 2a). In the cubic (100) single-crystalline silicon (SCS), Young's modulus is 90° symmetric[19], resulting in the same symmetry between the two m = 3 modes. However, due to the asymmetry in other directions of SCS lattice, the modal displacements at each of the anti-nodes are different, resulting in a net displacement at the central node, where the resonator will be anchored (Fig. 2b, c). Such displacement will cause a strain energy coupling between the resonator and substrate, leading to the acoustic energy dissipating into the handle layer and reducing anchor Q. More importantly, the substrate coupling compromises the mode degeneracy. Process variations like anchor misalignment and anchor geometrical imperfection can introduce excessive stiffness and damping cross-coupling between the two m = 3 BAW modes; in addition, mounting conditions and PCB stress could also affect the resonator through such substrate coupling. Contrary to the cubic (100) Si, 4H-SiC possesses hexagonal crystal symmetry[20]. The two m = 3 modes have uniform modal displace at all the anti-nodes with a strain equilibrium at the center anchor with near zero coupling to the substrate. With the addition of a folded beam decoupling network, the simulated anchor Q, using finite element analysis (FEA) in COMSOL is greater than 2 ×10⁹ across all process parameters, resulting in a fully acoustically-decoupled design from the underlying substrate.

The upper limit of achievable Q is bounded by the intrinsic Akhiezer dissipation, which is due to the interaction between phonons and acoustic waves, and leads to the reduction of lattice thermal vibrations and the suppression of energy dissipation in a material. The numerical expression for the frequency Q (fQ) product of an Akhiezer damping limited mechanical resonator has been studied in refs. 15,21:

$$f \cdot Q = \frac{\rho \cdot c^2 \cdot c_D^2}{2\pi \cdot \gamma^2 \cdot k \cdot T} \qquad (1)$$

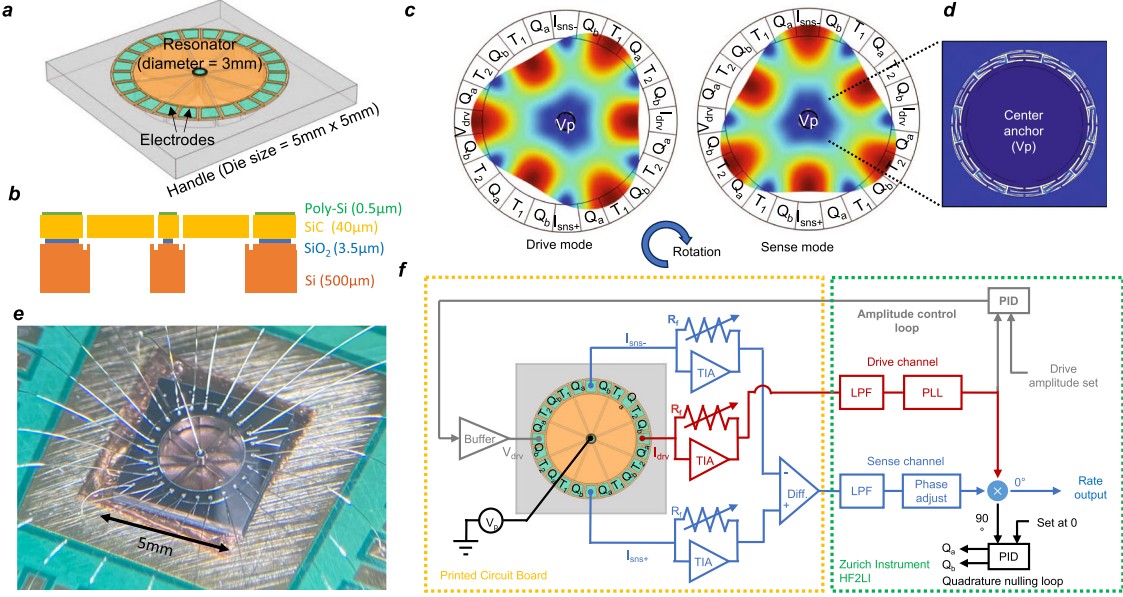

**Fig. 1 | Bulk acoustic wave disk gyroscope in 4H-silicon carbide. a** Resonant gyroscope design on the silicon carbide-on-insulator platform and **b** its cross-section illustration with thicknesses of each layer. **c** The operational mode-shapes, the gyroscopic m = 3 bulk acoustic wave modes are coupled to each other through the Coriolis effect. The electrode layout is designed accordingly, where $V_{drv}$ is the driving electrode; $I_{drv}$ is the drive mode current output electrode; $I_{sns+/-}$ are the sense mode differential current output electrode; $T_{1/2}$ and $Q_{a/b}$ are the two frequency tuning and quadrature nulling electrodes, respectively; $V_p$ is the polarization electrode. **d** The inset shows the deformation in the decoupling folded beam to isolate the acoustic wave from the central anchor. **e** Optical photograph of the fabricated device wire-bonded onto a printed circuit board (reproduced from ref. 16), the silicon carbide resonator is transparent and can be seen through in the optical picture. **f** The interface electronic architecture for the silicon carbide gyroscope, the abbreviations stand for the following: TIA transimpedance amplifier, LPF low pass filter, PLL phase lock loop, PID Proportional Integral Derivative controller, Diff. differential amplifier, $R_f$ feedback resistor.

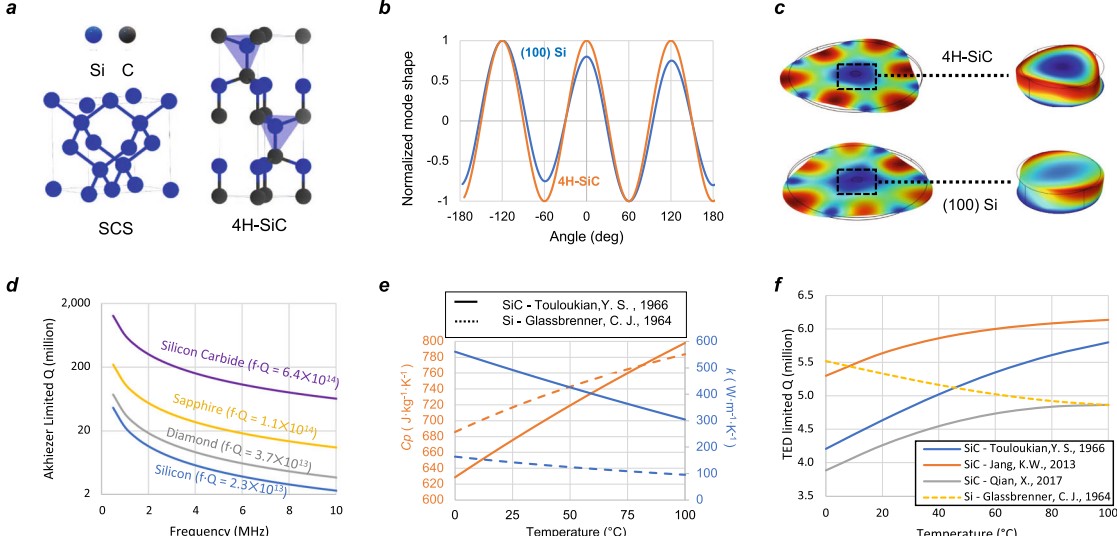

**Fig. 2 | Comparison between silicon and silicon carbide. a** The cubic stricture single crystalline silicon (SCS) lattice and hexagonal lattice in 4H-silicon carbide. **b** The displacement amplitude at the edge of the resonator of the m = 3 mode in single crystalline silicon and 4H-SiC. **c** The m = 3 mode in (100) silicon and 4H-silicon carbide, with a zoom-in view to show the uniformity difference in the net displacement at the center anchoring point. **d** The Akhiezer limit in silicon carbide, silicon, and other common materials. **e** The temperature-dependent specific heat $Cp$ and thermal conductivity $k$ value in silicon and silicon carbide from reference[22,25]. **f** With thermal properties from different references[22,25,28,48], the simulated thermal elastic damping (TED) limited $Q$ vs. temperature of the similar design in silicon and 4H-silicon carbide

Where $\rho$ is the material density; $c$ is the acoustic wave velocity, $c_D$ is the Debye average velocity; $\gamma$ is the Grüneisen parameter; $k$ is the thermal conductivity; and $T$ is the temperature.

From Eq. (1), we see that the Akhiezer damping solely depends on the material properties regardless of the resonator geometry. Compared to silicon, silicon carbide possesses higher $c_D$ and a lower $\gamma$, effectively preventing energy transfer through normal and Umklapp scattering when acoustic waves perturb the equilibrium distribution of phonons. Consequentially, the Akhiezer limit in silicon carbide is 30× higher than that in silicon and remains outstanding in other common or novel semiconductor materials, as shown in Fig. 2d.

Another intrinsic loss factor that often dominates the overall $Q$ in a MEMS resonator is thermo-elastic damping (TED). For the disk gyroscope design in this article, using 4H-SiC material properties from ref. 22, the simulated TED $Q$ is 4.6 million at room temperature. A higher $Q$ can be achieved in a completely solid disk resonator without a central decoupling beam network, as reported in ref. 23 where an alternative substrate decoupling design using phononic crystal (PnC) in the handle layer eliminates the need for a folded beam and shows an ultra-high $Q$ of 18 million in a solid 4H-SiC disk resonator. However, the PnC design requires very high precision in geometrical definition and excessive footprint, and as such it is not adopted in this work. Additionally, previous work has shown that the achievable $Q$ factor in a capacitive SiC disk resonator of similar dimension, with or without a decoupling network, is mainly limited by surface roughhouses on the side wall, loading the typical measured TED 1.5~3 million, which is still considerably higher than that of silicon counterparts at the megahertz range[23,24].

### Positive temperature coefficient of QTED (TCQ)

Interestingly, the material properties in SiC are strongly temperature-dependent in a way that benefits the $Q$ at a higher temperature. Particularly, the heat capacity $C_p$ and thermal conductivity $k$ have a much larger rate of change in 4H-SiC than in silicon for the same temperature range[22,25], as shown in Fig. 2e. Unfortunately, despite a few studies have reported the temperature-dependent thermal property in 4H-SiC, the reported values in each reference have large variations[22,26–30]. These variations arise from different doping types and concentrations[28], pressure[29], and sample

preparation methods, such as thin film[30] versus bulk wafers[28]. Most of the references deal with SiC for power electronic applications and very limited resources quantifying 4H-SiC as an acoustic material for MEMS are available. Still using the temperature-dependent material properties from either one of these references in the TED multiphysics simulation would yield a positive TCQ, in contrast to the negative TCQ of silicon devices of the same design as shown in Fig. 2f. Ovenization of a resonator at its highest operating temperature is a technique often used in timing resonators to stabilize temperature and achieve high stability[31], but rarely implemented in a silicon MEMS gyroscope due to the negative silicon TCQ degrading the gyroscope performance at elevated temperatures. A SiC resonator with opposite TCQ can benefit from ovenization to improve its stability against an unstable operational environment.

### DRIE capacitive transduction gaps

The structure and the parallel capacitive gap of the resonant gyroscope are defined by wafer-level SiC deep reactive ion etching (DRIE) Fig. 3a shows a picture of the author holding a processed 4" SiCOI wafer with various resonators designs on it. Figure 3b shows the bird-view scanning electron microscopy (SEM) of the BAW disk gyroscope on the SiCOI wafer. The zoom-in SEM of the decoupling network and peripheral capacitive electrodes in Fig. 3c, d shows a pristine etching profile with smooth sidewalls to suppress TED due to surface roughness. Unlike the BOSCH process used for silicon DRIE where the etching and passivation cycles alternate to achieve an anisotropic trench profile[32], SiC DRIE is self-passivated and relies on high-power plasma etching using a hard mask such as an electroplated Nickel[33–35]. As such the aspect ratio of SiC DRIE today is limited to 1:10 ~ 1:15. In this work, SiC-on-insulator wafers with 40-µm thick device layer were chosen and targeting a trench width of 3.5 µm (Fig. 3e). The detailed SiC DRIE recipe is shown in supplementary Fig. 1 with Supplementary Notes 1 to highlight the key parameters to improve the etching profile.

Although this capacitive gap size is relatively large, the high $Q$ factor can sufficiently reduce the motional impedance. And thus, even with a moderate gap size, a 4H-SiC BAW resonator can be effectively actuated electrostatically. In addition, large DC voltages can be applied for polarization, which further guarantees the low motional impedance and high

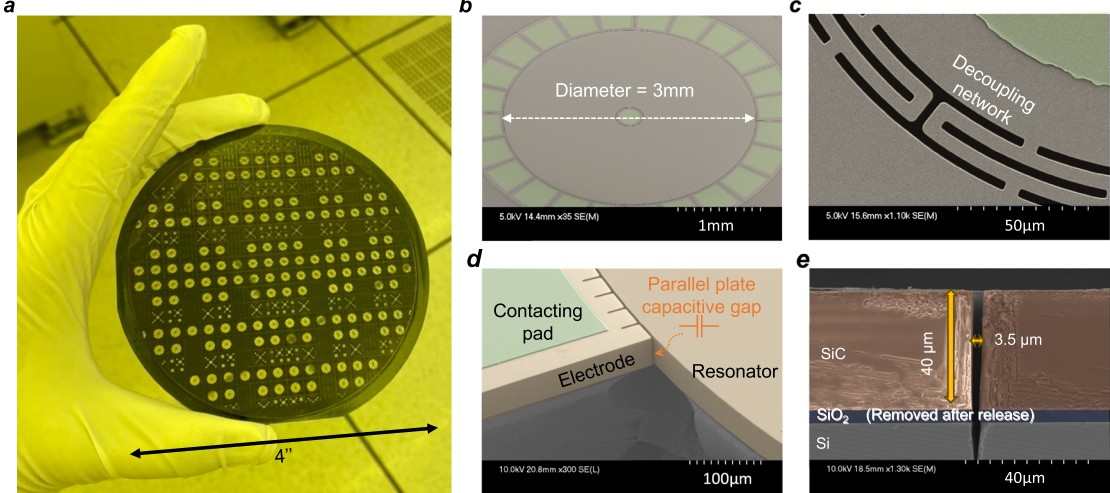

**Fig. 3 | Processed silicon carbide-on-insulator wafer and devices.** Figure reproduced from ref. 16. **a** A process 4" silicon carbide-on-insulator wafer with various resonator designs, some of them have backside openings in the silicon handle layer, which can be seen through from the transparent silicon carbide device layer on top. **b** Bird view scanning electron microscopy showing the 3 mm silicon carbide disk resonant gyroscope. **c, a** Zoomed-in view of the decoupling network at the resonator center. **d** Details of the capacitive trench between the electrode and the resonator, taken by cleaving off one of the adjacent electrodes. **e** Cross-section view of the deep reactive iron etched trench profile, taken by dicing through the resonator; the microscopy is colored to show different layers, and the buried oxide $SiO_2$ layer is removed during the releasing process.

electrical scale factor. Another challenge for high-stiffness BAW gyroscopes in using a wide capacitive gap is the effective tuning range. The relation between the tuned frequency $\omega_{tuned}$ and the gap size is given by:

$$\omega_{tuned} = \omega_0 \sqrt{1 - V_P^2 \frac{\varepsilon \cdot A}{kg^3}} \qquad (2)$$

Where $\omega_0$ is the mode-matched center frequency; $\varepsilon$ is permittivity constant; $V_P$ is the DC polarization and tuning voltage; $A$ and $g$ are the parallel electrode area and gap size; and $k$ is the effective mechanical stiffness of the resonator. Compared with silicon BAW gyroscopes, which typically need a sub-micron capacitive gap to enable larger electrostatic tuning and compensate for larger gyroscopic mode frequency splits as well as electromechanical coupling due to their limited $Q$ factor[12,13,18], the 4H-SiC BAW gyroscope can work with a larger transduction gaps since the as-born frequency split is very small and the operating $Q$ is very high. In this work, devices with a 3.5 µm gap were successfully mode-matched and operated owing to the hexagonal in-plane isotropic lattice of 4H-SiC, which resulted in the as-born frequency split being only a few ppm and falling within the electrostatic tuning range.

## Frequency response measurement

Numerous 4H-SiC disk resonators across two 4" SiC-on-insulator wafers were characterized at room temperature in a vacuum chamber, with the statistical data visualized in Fig. 4a–d. The measured frequency responses are very consistent at a center frequency of 3.011 MHz with $Q$ above 1.5 million at room temperature. The small as-born frequency with an average of 7.6ppm is beyond the reach of similar silicon devices. The discrepancy in the $Q$-factor between the two m = 3 modes is well constrained within 8%, minimizing the in-phase error in the gyroscope output. With a high polarization DC voltage of 260 V, the motional impedance is 216 kΩ and can be mode-matched via electrostatic spring softening. Figure 4e, f shows mode matching and alignment of device #10 with an as-born frequency split of 9.3 ppm by applying a large negative DC voltage at the tuning electrode. The indirect response takes the differential readout from the sense mode, and upon mode matching, over 45 dB cross-mode isolation was achieved. The same device is then ovenized at 80 °C, measuring an increased $Q$ of 4.6 million, corresponding to

76 kΩ motional impedance. Without the need to re-adjust the tuning voltage set at room temperature, the device remains mode-matched after ovenization, as shown in Fig. 4g.

Two devices are measured across a temperature range of 0~100 °C. Figure 4h, i shows the corresponding frequency and $Q$ value. A linear TCF of −14.3 ppm·°C$^{-1}$ was extracted from the data, in agreement with the literature[20,24]. The $Q$ increased from 0 to 70 °C and plateaued above 80 °C; such positive TCQ for 4H-SiC is as expected and the values are close to the TED simulation results, although the values are not a perfect match due to the uncertainty of the exact value of the actual material properties. It is worth noting that despite elevated temperature may change the ambient pressure of a hermetically sealed resonator and thus affect its $Q$ through air damping, in this work, vacuum is achieved via a turbo pump to maintain a constant pressure level of $10^{-5}$ mTorr, to rule out the squeeze film effect. In addition to the m = 3 BAW disk gyroscopes, other BAW resonators from this wafer, such as square Lamé mode resonators, also show a similar consistent temperature response of frequency and $Q$, the characterization of which can be found in the Supplementary Fig. 2 and Note 2, proving that such uncommon positive $TCQ_{TED}$ for BAW resonators is due to the material properties of SiC.

To characterize the resonator linearity, which feeds into how hard the resonator can be driven to increase the Coriolis coupling, forward-and-backward frequency sweep characterization was conducted with different driving power using a vector network analyzer (VNA), the results of which are shown in Fig. 4j. Spring softening non-linearity was observed in the resonator starting with an input power of 5 dBm, where the backward sweep reached a higher output than the forward sweep, followed by a sudden drop in magnitude at lower frequency. As power increases further, the resonator shows a more pronounced Duffing effect, with an increased frequency bifurcation between forward and backward sweep. Such non-linearity is a result of the large displacement amplitude in a parallel plate capacitive transducer. In gyroscope operation, the largest driving voltage used to actuate the resonator is kept at 0 dBm or 650 mV peak-to-peak, resulting in a disk vibration amplitude of 160 nm as obtained from FEA simulation, which corresponds to 5% of the gap size. Stronger actuation of the resonator will push the vibration amplitude to exceed the linear range in a parallel plate capacitive transducer, not only causing instability in the frequency spectrum but also adding noise to the gyroscope output through high-order flicker noise up-conversion.

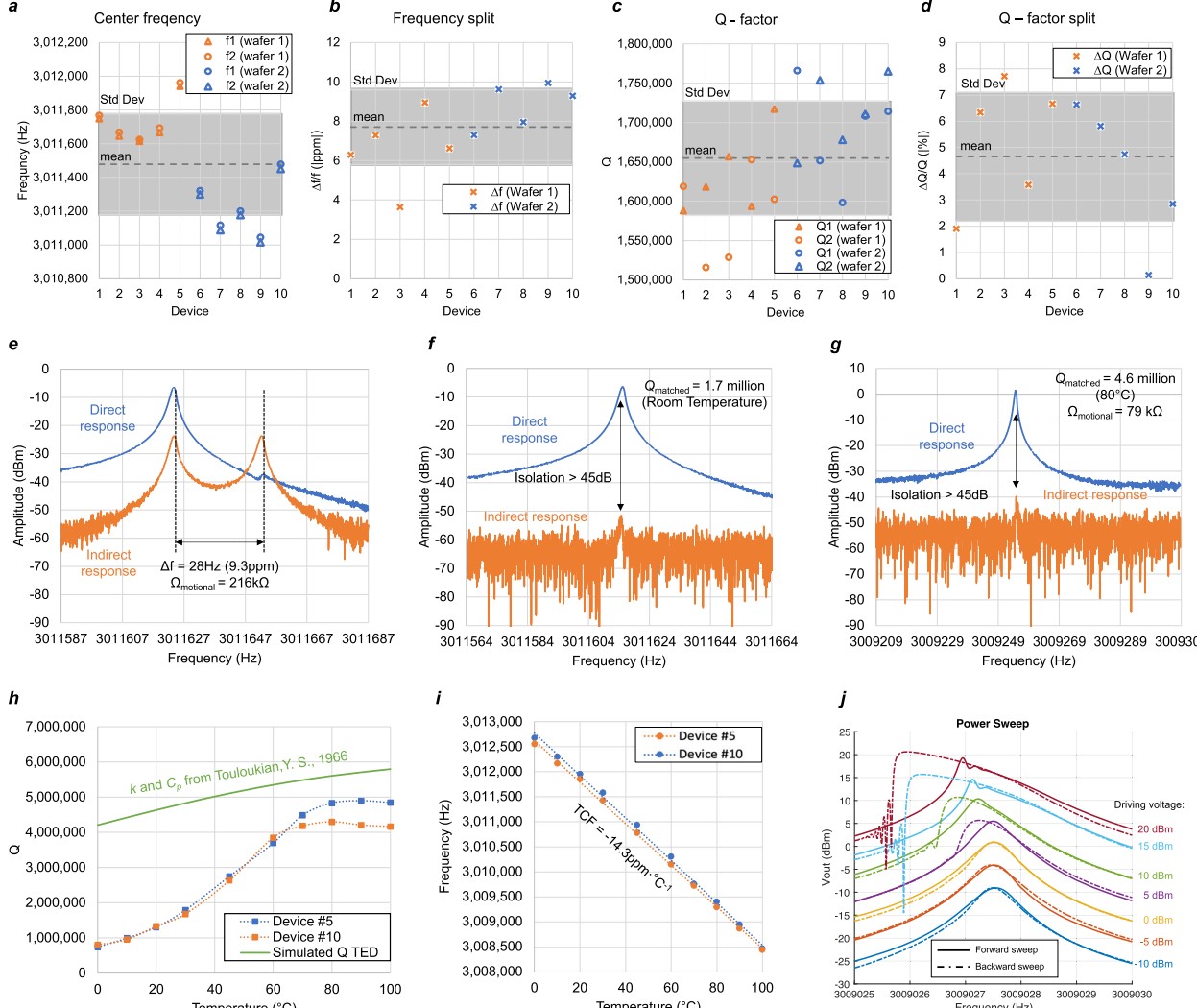

**Fig. 4 | Frequency response measurement. a** The statistical data of the center frequency: f1, f2 of each m = 3 mode, **b** frequency split Δf, **c** Q-factor: Q1,Q2 of each mode and **d** Q-factor split, ΔQ, measured in 10 identical m-3 bulk acoustic wave (BAW) disk gyroscopes randomly selected from two SiCOI wafers. **e** The detailed frequency response of device #10, using data reproduced from ref. 16, the direct response is the drive mode output, and the indirect response is the differential readout from two sense mode output. the as-born device has a frequency split of 9.3 ppm, and the motional impedance ($\Omega_{motional}$) was measured to be 216 kΩ, **f** upon mode matched, the two frequency overlaps and the cross-mode isolation is above 45 dB, with a mode-matched Q ($Q_{matched}$) of 1.7 million at room temperature. **g** The ovenized resonator shows a mode-matched Q of 4.6 million at 80 °C, and cross-

mode isolation remains above 45 dB, corresponding to lower $\Omega_{motional}$ of only 79 kΩ. **h** A positive temperature of Q (TCQ) was measured from 0 to 70 °C, and plateaued toward above 80 °C, the positive TCQ in silicon carbide, in contrast to negative TCQ in silicon devices, is as expected in the simulation using thermal conductivity k and specific heat Cp from literature. **i** The resonators have a linear temperature coefficient of frequency (TCF) of −14.3 ppm·°C⁻¹. **j** Linearity characterization of a (not mode-matched) SiC BAW disk resonator, at 80 °C the resonator output remains linear until the driving voltage is beyond 0 dBm (~0.65 Vpp), which corresponds to a maximum linear displacement amplitude of 160 nm, above which, softening effect starts to appear.

## Gyroscope characterization

The rate response was recorded over 5 min for the 80 °C ovenized 4H-SiC gyroscope after mode matching, as shown in Fig. 5a. No discernable drift was observed and a scale factor of 215 nA·°⁻¹·s, with only 5% non-linearity error, was extracted for the device (Fig. 5b). The gyroscope characterization results in different temperatures are shown in Fig. 5c. The driving amplitude was kept at 160 nm, close to 5% of the gap size, to ensure linear capacitive transduction across all temperatures by adjusting the driving voltage with respect to the change in Q. The best ARW measured was 0.005°·h⁻¹/² when ovenized at 80 °C set by the operational temperature of the interfacing IC, and the BI achieved was consistently below 0.5°·h⁻¹ in all temperature ranges.

Compared with silicon BAW gyroscopes which often have 250 nanometer capacitive gap, the reported SiC device has a DRIE gap size that is ~10× larger[11–13,18,36]. Such a large gap is unfavorable for the tuning range and

high DC voltage requirement, it could also affect the noise performance. While the gap size limits the linear displacement range in the capacitive parallel plate, it is arguable that a 4H-SiC BAW gyroscope can benefit from a moderate gap to reach a higher mechanical scale factor with a maximum displacement amplitude near a few hundred nanometers. The ARW is comprised of the mechanical and electrical noise equivalent rates (i.e. MNEΩ and ENEΩ, respectively), given by:

$$ARW = \sqrt{MNE\Omega^2 + ENE\Omega^2} \quad (3)$$

$$MNE\Omega = \frac{1}{2 \cdot \lambda \cdot |q|} \sqrt{\frac{4k_B \cdot T}{\omega_0 \cdot m \cdot Q}} \quad (4)$$

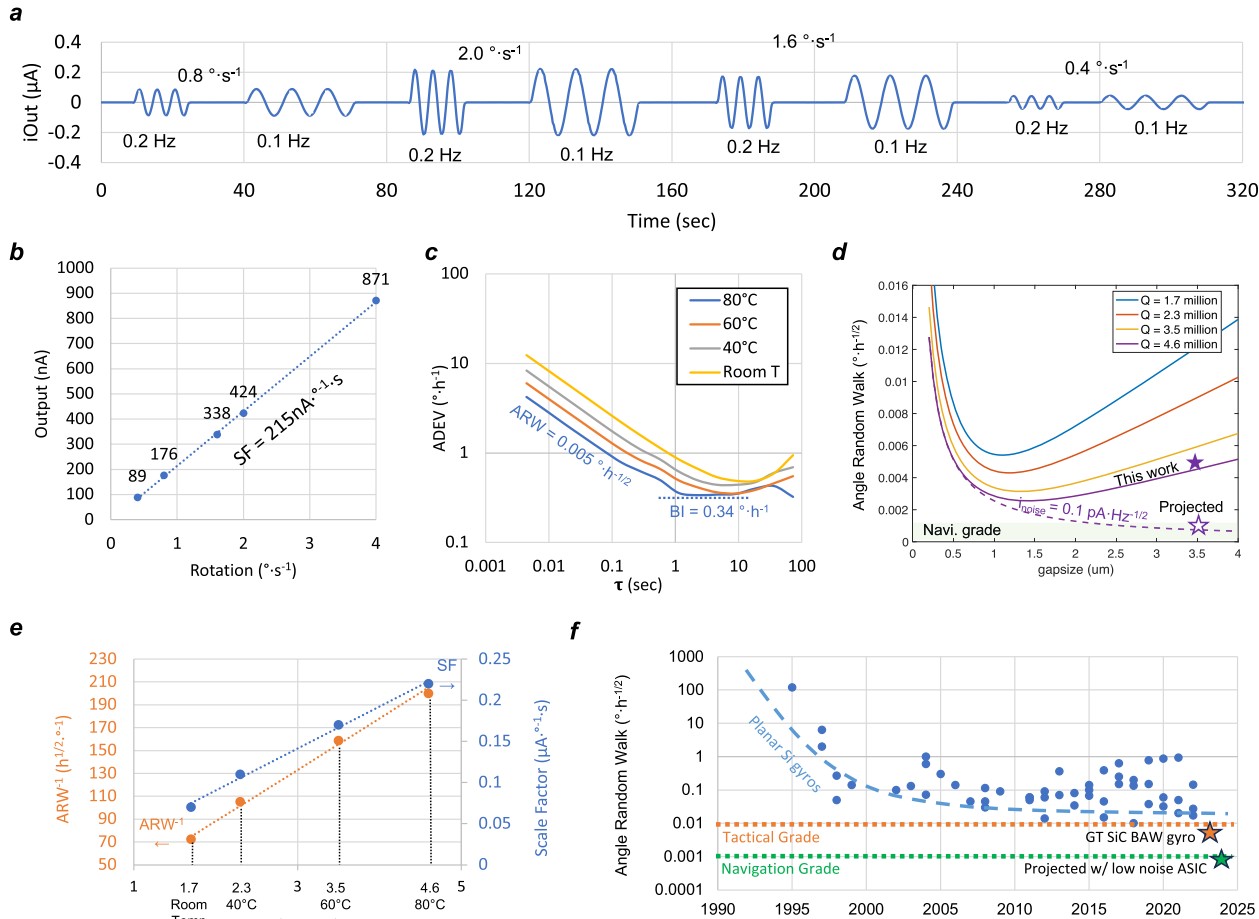

**Fig. 5 | 4H-silicon carbide gyroscope characterization. a** The rate responses of the ovenized gyroscope recorded over 5 min, plotted using data reproduced from ref. 16. **b** A scale factor (SF) of 215·°⁻¹·s with 5% non-linearity was extracted. **c** Allan deviation is measured across temperature when the device is maximally driven at 160 nm amplitude. **d** Calculated angle random walk (ARW) and bias instability (BI) vs. gap size assuming the drive mode displacement is 5% of the gap size, the input-referred noise ($i_{noise}$) is 3pA·Hz$^{-1/2}$. **e** The measured scale factor and inverse of ARW vs. Q-factor at different temperatures. **f** History of planer silicon gyroscope, which is facing a barrier near the tactical grade. This work promotes monocrystalline 4H-silicon carbide as an alternative platform for precision micromachined gyroscopes.

$$\text{ENE}\Omega = \frac{i_{\text{noise}}}{2\lambda \cdot Q \cdot |q| \cdot \varepsilon \cdot A \cdot V_p} \cdot g^2 \tag{5}$$

Where $\lambda$ is the Coriolis coupling coefficient between the two gyroscopic modes; $|q|$ is the drive mode vibration amplitude; $k_B$ is the Boltzmann constant; $T$ is the temperature in Kelvin; $i_{\text{noise}}$ is the input-referred noise of the interface electronics.

Both MNE$\Omega$ and ENE$\Omega$ can be scaled down by increasing the driving amplitude $|q|$, but $|q|$ must be small enough compared to the gap size (5% in this work) to ensure linear transduction. Therefore, the optimal gap size is facing a trade-off between the MNE$\Omega$ and ENE$\Omega$. With a typical input-referred current noise of 3 pA·Hz$^{-1/2}$ for our discrete electronics setup, one can plot the expected ARW vs. gap size for a 3 mm 4H-SiC disk resonator (Fig. 5d). The measured scale factor and inverse of ARW with different $Q$ were summarized in Fig. 5e and can be used to analyze noise contributions in this 4H-SiC BAW disk gyroscope. From Eqs. (4) and (5), Brownian noise induced MNE$\Omega$ is inversely proportional to $Q^{1/2}$ while the inverse of ENE$\Omega$ is linearly proportional to $Q$; considering that a linear fit best describes the inverse of ARW vs $Q$ change, we can conclude that the electronic noise is the main contributor, in agreement with our analytical derivations in Fig. 5d. Therefore, in order to break through the navigation requirement, the primary task is to reduce the input-referred current noise $i_{noise}$ of interface electronics. When $i_{noise}$ is reduced to 0.1 pA·Hz$^{-1/2}$, which is challenging but achievable with a dedicated application-specific integrated circuit

(ASIC)[37–39], the ARW is projected to be around $0.8 \times 10^{-3}$ °·h$^{-1/2}$ with everything else unchanged. However, the measured $Q$ in this design is still lower than the Akhiezer limit of SiC, considering that $Q$ of 20 million has been previously reported for a 6 MHz 4H-SiC lame resonator in ref. 40. Future work with an improved $Q$, along with a thicker substrate and optimized gap size, can eventually bring the performance of the SiC bulk acoustic wave gyroscope to better than $0.1 \times 10^{-3}$ °·h$^{-1/2}$, exceeding that of the best optical ring laser gyroscopes (RLG).

The gyroscope design and performance matrix are summarized in Table 1. While the ARW and BI fall between the tactical and navigation grade, the demonstrated bandwidth need to be further improved. The SiC disk gyroscopes in this work are operated in an open loop mode, with the operational bandwidth being half the 3 dB bandwidth of the resonator. A closed-loop force-to-rebalance architecture can expand the operational bandwidth by 100× without degrading the noise performance to meet the bandwidth requirement[41,42].

Reviewing the history of planer MEMS gyroscopes, much improvement in the ARW of silicon devices has been made during the 1990s' shortly after the MEMS gyroscope was first introduced by Draper Lab[43]. However, over the past two decades, the ARW has remained relatively stable at a comparable level between 1°·h$^{-1/2}$ to 0.1°·h$^{-1/2}$, as shown in Fig. 5f, with the list of references being provided in the Supplementary Table 1 and reference. Alternatively, 3D wine-glass gyroscopes have been developed to go beyond planar devices, achieving very low ARW and BI by increasing the effective mass and $Q > 1$ million, such as the fused silica or fused quartz hemisphere

**Table 1 | The 4H-silicon carbide bulk acoustic wave disk gyroscope performance matrix**

| Parameter | Value |
|---|---|
| Size | $5 \times 5 \times 0.5$ mm$^3$ |
| Center frequency | 3.011 MHz |
| Polarization voltage | 260 V |
| Ovenization | 80 °C |
| $Q$ | $4.6 \times 10^6$ |
| Bandwidth | 0.32 Hz (open-loop) |
| Driving voltage | 650 mV |
| Scale factor | 215 nA·$^\circ$$^{-1}$·s |
| Angle random walk | 0.005°·h$^{-1/2}$ |
| Bias instability | 0.34°h$^{-1}$ |

resonant gyroscope (HRG), based on a non-traditional "glass-blowing" process[44,45]. However, these designs operate at low kilohertz frequency range, which makes the 3 dB bandwidth and full-scale range very limited due to their ultra-high $Q$ and low frequency. On the contrary, the 4H-SiC BAW disk gyroscope in this work operates at the megahertz range, providing nearly 3 orders of magnitude expansion in the bandwidth and full-scale range, with comparable ARW and BI. In addition, the BAW gyroscope is much smaller in size and substantially insensitive to linear shock and vibration in the environment owing to its very stiff structure.

## Conclusions

In this work, we present a 4H-SiC BAW disk resonant gyroscope, for the first time, to create a leap in the performance of MEMS gyroscopes. The presented gyroscope has an ovenized $Q$ of 4.6 million at 80 °C, angle random walk of 0.005°·h$^{-1/2}$, and sub-degree per hour bias instability limited by interface electronics. The unique thermal properties in SiC allow the TED-limited resonator to achieve higher $Q$ at elevated temperatures, ensuring better performances when ovenized. Devices are fabricated on bonded 40-μm thick SiC-on-insulator 4" wafers using a wafer-level process. The 3.5 μm wide capacitive trenches are defined by high-power SiC DRIE. Despite this gap size being unfavorably large for capacitive actuation and tuning, the high $Q$ factor in the 4H-SiC disk resonator can ensure a low motional impedance for effective transduction, and the hexagonal in-plane isotropic crystal symmetry guarantees a small as-born frequency split between the two gyroscopic m = 3 BAW modes, allowing complete mode matching via electrostatic tuning. The 3pA·Hz$^{-1/2}$ input referred noise is the major limiting factor for the reported gyroscope, by designing an ultra-low noise ASIC, with a force-to-rebalance closed loop to expand the operational bandwidth, the presented gyroscope is projected to break through the long-term inertial navigation requirement. The large polarization DC voltage in this work is due to the wide DRIE-defined capacitive gap. To reduce the tuning and bias voltage to below 50 V, smaller gap size of 1–2 μm can be used. Advances in the SiC DRIE or alternative techniques such as the high aspect ratio combined polysilicon and single crystalline silicon (HARPSS) process[36,46,47] may be adapted to the SiCOI platform for creating a narrower capacitive gap.

## Methods
### Finite element analysis
COMSOL Multiphysics® was used for finite element analysis. The disk resonator is reconstructed in 3D based on the actual device measurement along with some imperfection assumptions, and the material properties are adopted from references[20,22,25–29,48]. The meshing conditions and results can be found in the supplementary Fig. 3. The following simulations are performed in this work:

Anchor loss is simulated using a solid mechanics model. A perfect match layer with critical dimensions greater than a quarter wavelength is put around the 500 μm thick handle layer. A fixed constraint is placed at the edge of the perfect match layer with everything else being free and static initial condition. An eigenfrequency study with geometric non-linearity is performed to find the acoustic energy lost through the perfect match layer.

Thermo-elastic damping (TED) simulation used the thermal expansion Multiphysics model coupling heat transfer and solid mechanics. The handle layer is omitted in this model with a fixed constraint applied to the bottom silicon oxide at the center of the resonator. Temperature with harmonic perturbation boundary condition is applied to the same location and the thermal insulator is applied to all other external surfaces. A pre-stressed eigenfrequency study is used to calculate the energy dissipation due to thermal perturbation.

Electrostatic tuning and actuation is studied using electromechanical force coupling solid mechanics to electrostatics models. The handle layer is also omitted with a fixed boundary condition applied to the center silicon oxide post. Vp is applied as the domain constraint to the disk resonator. The electrodes are modeled as 3.5 μm air gaps with proper voltage boundary conditions. For the driving electrode, the AC is applied as a harmonic perturbation. All air gaps are applied with moving mesh to account for the large deformation. The tuning range is found by a prestressed eigenfrequency study by parameter sweep of the tuning voltage, and a prestressed frequency domain study is performed to analyze the displacement amplitude.

### SiC BAW disk resonator fabrication
The devices in this work are fabricated on 4" SiC-on-insulator wafers. SiCOI wafers are not commercially available, and are fabricated using the following process flow: a 500-μm thick n-type 4H-SiC wafer, with resistivity between 0.015 and 0.028 Ω·cm, purchased from Wolfspeed Inc., was first deposited with 3.5 μm TEOS oxide and then bonded to a bare silicon handle wafer using Si-SiO$_2$ bond after kiss polishing. Then the SiC layer was ground down to the target thickness, followed by another kiss polishing and chemical-mechanical polishing to achieve a low surface roughness below 5 Å. To fabricate the devices on the SiCOI wafer, first, a thin layer of phosphorus-doped polycrystalline silicon with resistivity around 0.01Ω·cm, was deposited and patterned to define the electrode contact pads. Then, an electroplated nickel mask was patterned and used for SiC etching. The etching was done using a Plasma-Therm Versaline® etcher. Next, the release holes in the silicon handle layer are etched through using the standard BOSCH process. After that, the wafer is diced into individual dies, followed by annealing at 1100 °C to strengthen the oxide-silicon bonding and short rapid thermal annealing at 1200 °C to form an ohmic contact between the polycrystalline silicon electrode pad and with the wide-band SiC device layer. Finally, the device is released in hydrofluoric acid. The detailed process flow can be found in Supplementary Fig. 4.

### Frequency response measurement and gyroscope testing setup
The gyroscopes are wirebonded on a laboratory PCB with a unity gain buffer to drive the resonator driving electrode. Three TIAs with 100 kΩ gain are connected to the current output electrodes. The sense modes outputs have an additional differential stage with unit gain before being connected to the testing equipment. The buffer, differentiator, and TIA used operational amplifiers OPA656 and OPA657 from Texas Instruments, Inc., with gain bandwidth products of 500 MHz and 1.6 GHz, respectively.

The frequency response was measured using a Keysight E5080A Network Analyzer; during each frequency spectrum sweep, a total of 1601 data points was measured with an intermediate frequency bandwidth (IFBW) set at 20 Hz. The gyroscopes characterization was performed using a Zurich HF2LI lock-in amplifier, to create an oscillation loop at the drive mode frequency. After synchronous demodulation of the sense mode signal, closed-loop quadrature nulling was implemented. The gyroscope is capped on the PCB, with an external pump to reach a vacuum level of $5 \times 10^{-3}$ mbar. The PCB is fixed on the rate table within an environmental chamber to heat

up both the resonator and the circuity. The detailed testing setup configuration and conditions are shown the Supplementary Fig. 5 and Note 3.

## Data availability
The data that support the plots within this paper and other findings of this study are available from the corresponding authors upon reasonable request.

## Code availability
The code that supports the theoretical calculation within this paper and other findings of this study are available from the corresponding authors upon reasonable request.

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

## Acknowledgements

The research was sponsored by the Army Research Laboratory with SEMI-PNT and was accomplished under Cooperative Agreement Number W911NF-22-2-0050. The views and conclusions contained in this document are those of the authors and should not be interpreted as representing the official policies, either expressed or implied, of the Army Research Laboratory or the U.S. Government or SEMI. The U.S. Government is authorized to reproduce and distribute reprints for Government purposes notwithstanding any copyright notation herein. We thank Plasma Therm Inc. for their assistance in SiC DRIE.

## Author contributions

Z.L. designed, fabricated, and characterized the 4H-SiC BAW disk gyroscope and wrote the manuscript. Y.L. helped with the design and characterization. C.W. helped with device fabrication. H.W. helped with device characterization and TCQ analysis. F.A. led the project, provided guidance and funding, and helped with manuscript writing.

## Competing interests

Z.L., H.W. and F.A. are partially or fully associated with StethX Microsystems Inc., a company interested in commercializing silicon carbide-based micro-electro-mechanical system devices.
