## [Peer Review File · Communications Engineering]

Reviewers' comments:

Reviewer #1 (Remarks to the Author):

This paper for the first time reported a chip-scale Coriolis gyroscope based on the SiC material. Compared with conventional silicon material, SiC has the potential to provide an even higher mechanical quality factor, because it can provide a much lower Akhiezer dissipation limit and more balanced mode shapes for the $m=3$ operation modes. As a long-anticipated research, this paper reported exciting results. The measured quality factor of 4.6 million at an 80°C ovenized environment is very appealing. An open-loop gyroscope prototype providing an angle random walk of 0.005°/√h and a bias instability of 0.34°/h is also very promising. There is no doubt that this is a successful exploration which directs a new approach to developing high-performance MEMS gyroscopes and other MEMS resonant devices.

The originality and significance of this work are superb, and supported by solid data. I strongly recommend its publication, but only after the following issues are addressed.

1. The manuscript states that the disk resonators are operated in a pair of $m=3$ elliptical modes. However, the mode shapes of the $m=3$ modes are obviously not ellipses. Only the $m=2$ modes are elliptical.
2. The paper discussed the influence of the temperature on the quality factor. However, pressure is also a very important factor that may affect the quality factor. I think the authors should provide some data-supported discussion about this, which I suspect might be helpful in explaining the quality-factor plateau in Figure 4d.
3. The authors may consider visualizing the data in Table 1 for better clarity. Also, the quality factor difference is very important for the gyroscope performance because it may provide in-phase errors in the gyroscope output.
4. The linear to nonlinear frequency response characterization in Figure 4f is not very clear. The authors may consider conducting a forward-and-backwards sweeping characterization to better illustrate the nonlinear bifurcation.

Reviewer #2 (Remarks to the Author):

The authors report on on-chip gyroscope devices using 4H-SiC. They theoretically and quantitatively describe the anticipated benefits of utilizing the superior properties of 4H-SiC. Following this, they fabricate actual gyroscope devices, assessing its resonant characteristics and performance. They

successfully demonstrates the advantages of 4H-SiC gyroscopes over traditional Si gyroscopes. It is very interesting heating this device at 80degreeC gives us superior properties.

For long time, some researchers have studied SiC MEMS devices, most of them were for harsh environment MEMS devices. This paper proposed the new direction of 4H-SiC MEMS. This paper is very valuable.

However, it is believed that several revisions are necessary before the paper is suitable for publication.

1) previous study

Very high-Q factors was reported in single crystalline 4H-SiC cantilevers experimentally over 10 years ago.

<https://doi.org/10.1016/j.sna.2013.04.014>

The paper should be mentioned in introduction part as well as 4H-SiC BAW gyroscope part (around [20]-[22]).

2) missing information

a) diameter and size of chip in Fig. 1b. (3mm diameter and 4-5mm square?)

a) thickness of 4H-SiC (The value is shown in Fig. 3e. The value should be mentioned in the text as well as Fig. 1b.)

b) condition type and resistivity of 4H-SiC: it may be n-type low resistivity layer (around 30mΩcm?).

c) thickness and resistivity of poly-Si layer

3) In Fig. 5g, $0.001/h^{0.5}$ (a green star) is just expectation. It looks experimentally demonstrated. "Projected" should be added.

Reviewer #3 (Remarks to the Author):

The author has provided a comprehensive analysis of 4H silicon carbide bulk acoustic wave MEMS gyroscope with ultra-high Q, highlighting its significance in high-performance MEMS gyroscope design. However, there are a few areas that require further clarification.

On the 2nd page, the author mentions that substrate coupling compromises mode degeneracy ("More importantly, the substrate coupling compromises mode degeneracy."). It would be helpful if the author could provide a more detailed explanation of this phenomenon.

On the 3rd page, the author states that the achievable Q factor in a capacitive SiC disk resonator is mainly limited by surface roughness on the side wall ("Additionally, previous work has shown that the achievable Q factor in a capacitive SiC disk resonator of similar dimension, with or without a decoupling network, is mainly limited by surface roughness on the side wall..."). It would be beneficial if the author could summarize the specific etching techniques employed in this study to improve the smoothness of the SiC disk's side wall and subsequently enhance the Q factor.

In the caption of Figure 4, it is mentioned that a positive TCF was measured from 0~70°C, and plateaued above 80°C. However, it appears that there might be a typographical error, as it should likely be "a positive TQF" instead of "a positive TCF".

Figure 4(d) shows a softening effect and a decrease in resonant frequency with increasing actuation amplitude. It would be helpful if the author could provide an explanation for this unexpected behavior.

On the 5th page, the author states, "Considering that the inverse of ARW is linearly proportional to Q and noting that the $MNE\Omega$ and $ENE\Omega$ scale differently with respect to change in Q, we can conclude that the dominating noise is the electronic noise induced $ENE\Omega$, in agreement with our analytical derivations". It would be beneficial if the author could elaborate further on this point and provide a clearer explanation of their analytical derivations.

Rebuttal letter for

4H Silicon Carbide Bulk Acoustic Wave MEMS Gyroscope with Ultra-High Q for On-Chip Inertial Navigation

Zhenming Liu¹, Yaoyao Long¹, Charlotte Wehner¹, Haoran Wen², and Farrokh Ayazi^{1,2}

¹ School of Electrical and Computer Engineering, Georgia Institute of Technology, Atlanta, GA 30308, USA

² StethX Microsystems Inc., Atlanta, GA 30308, USA

Email: zhenming_liu@gatech.edu; ayazi@gatech.edu

Dear reviewers,

We are deeply grateful for the time and effort you have invested in providing constructive feedback on our initial submission.

In response, we have meticulously addressed each point. Enclosed with this letter, you will find our detailed responses to each comment. For ease of review, we have highlighted our responses in blue and marked the changes in the manuscript in red.

We believe that these revisions have significantly enhanced the quality and clarity of our work. We are hopeful that these changes shall meet your expectations and align well with the standards of Communication Engineering.

Thank you for your consideration and the opportunity to improve our manuscript.

Best regards,

Zhenming Liu, on behalf of all the authors

Graduate Research Assistant
School of Electrical and Computer Engineering
Georgia Institute of Technology
85 5th Street NW
Atlanta, Georgia 30308
+1-404-451-5796 (mobile)
zliu387@gatech.edu

Reviewers' comments:

Reviewer #1 (Remarks to the Author):

This paper for the first time reported a chip-scale Coriolis gyroscope based on the SiC material. Compared with conventional silicon material, SiC has the potential to provide an even higher mechanical quality factor, because it can provide a much lower Akhiezer dissipation limit and more balanced mode shapes for the $m=3$ operation modes. As a long-anticipated research, this paper reported exciting results. The measured quality factor of 4.6 million at an 80°C ovenized environment is very appealing. An open-loop gyroscope prototype providing an angle random walk of 0.005°/√h and a bias instability of 0.34°/h is also very promising. There is no doubt that this is a successful exploration which directs a new approach to developing high-performance MEMS gyroscopes and other MEMS resonant devices. The originality and significance of this work are superb, and supported by solid data. I strongly recommend its publication, but only after the following issues are addressed.

1. The manuscript states that the disk resonators are operated in a pair of $m=3$ elliptical modes. However, the mode shapes of the $m=3$ modes are obviously not ellipses. Only the $m=2$ modes are elliptical.

Thank you for the correction, all “ $m=3$ elliptical mode(s)” has been replaced with “ $m=3$ BAW mode(s)” to avoid confusion

2. The paper discussed the influence of the temperature on the quality factor. However, pressure is also a very important factor that may affect the quality factor. I think the authors should provide some data-supported discussion about this, which I suspect might be helpful in explaining the quality-factor plateau in Figure 4d.

Thank you for reminding us to consider the pressure effect on the quality factor. In this work, we used a turbo pump to maintain a constant pressure level. The following text was added to the manuscript for clarification:

... although the values are not a perfect match due to the uncertainty of the exact value of the actual material properties. **It is worth noting that despite elevated temperature may change the ambient pressure of a hermetically sealed resonator and thus affect its Q through air damping, in this work, vacuum is achieved via a turbo pump to maintain a constant pressure level of 10^{-5} mTorr, to rule out the squeeze film effect. In addition to the $m=3$ BAW disk gyroscopes, other BAW resonators from this wafer, such as square Lamé mode resonators, ...**

3. The authors may consider visualizing the data in Table 1 for better clarity. Also, the quality factor difference is very important for the gyroscope performance because it may provide in-phase errors in the gyroscope output.

Thank you for the suggestion. We replaced Table 1 with visualized data in Fig.4. We used a shaded area to show the standard deviation and a dashed line to indicate the mean value. We also highlighted that the Q mismatch in these resonators is less than 8%, which minimizes the in-phase error in the gyroscope output:

...were characterized at room temperature in a vacuum chamber, **with the statistical data visualized in Fig. 4a~d. The measured frequency responses are very consistent at a center frequency of 3.011MHz with Q above 1.5 million at room temperature. The small as-born frequency with an average of 7.6ppm is beyond the reach of similar silicon devices. The**

discrepancy in the Q -factor between the two $m=3$ modes is well constrained within 8%, minimizing the in-phase error in the gyroscope output. With a high polarization...

4. The linear to nonlinear frequency response characterization in Figure 4f is not very clear. The authors may consider conducting a forward-and-backwards sweeping characterization to better illustrate the nonlinear bifurcation.

Thank you for the suggestion, we have conducted the recommended forward-and-backward sweeping characterization and replaced the old plot. With backward sweeping, the duffing effect becomes more obvious. Beginning at 5dBm, the output from the backward sweep went higher than the forward sweep, followed by a sudden drop in the output. Such duffing is more pronounced as we further increase the drive power to 10,15, and 20dBm.

... To characterize the resonator linearity, which feeds into how hard the resonator can be driven to increase the Coriolis coupling, forward-and-backward frequency sweep characterization was conducted with different driving power using a vector network analyzer (VNA), the results of which are shown in Fig. 4j). Spring softening non-linearity was observed in the resonator starting with an input power of 5dBm, where the backward sweep reached a higher output than the forward sweep, followed by a sudden drop in magnitude at lower frequency. As power increases further, the resonator shows a more pronounced Duffing effect, with an increased frequency bifurcation between forward and backward sweep. Such non-linearity is a result of the large displacement amplitude in a parallel plate capacitive transducer. In gyroscope operation, the largest driving voltage used to actuate the resonator is kept at 0dBm or 650mV peak-to-peak, resulting in a disk vibration amplitude of 160nm as obtained from FEA simulation, which corresponds to 5% of the gap size. Stronger actuation of the resonator will push the vibration amplitude to exceed the linear range in a parallel plate capacitive transducer, not only causing instability in the frequency spectrum but also adding noise to the gyroscope output through high-order flicker noise up-conversion. ...

Reviewer #2 (Remarks to the Author):

The authors report on on-chip gyroscope devices using 4H-SiC. They theoretically and quantitatively describe the anticipated benefits of utilizing the superior properties of 4H-SiC. Following this, they fabricate actual gyroscope devices, assessing its resonant characteristics and performance. They successfully demonstrate the advantages of 4H-SiC gyroscopes over traditional Si gyroscopes. It is very interesting heating this device at 80°C gives us superior properties.

For long time, some researchers have studied SiC MEMS devices, most of them were for harsh environment MEMS devices. This paper proposed the new direction of 4H-SiC MEMS. This paper is very valuable.

However, it is believed that several revisions are necessary before the paper is suitable for publication.

1) previous study

Very high-Q factors was reported in single crystalline 4H-SiC cantilevers experimentally over 10 years ago.

<https://doi.org/10.1016/j.sna.2013.04.014>

The paper should be mentioned in introduction part as well as 4H-SiC BAW gyroscope part (around [20]-[22]).

Thank you for the reference recommendation; we added the suggested one (as reference [16]) to the 4H-SiC BAW gyro part (after the Introduction section) to introduce the thin film-type SiC devices and highlighted the Q is enhanced when the film quality improves. Then, we introduced our research as a different path by using the bulk SiC.

4H-SiC BAW gyroscope

... While SiC MEMS has drawn much attention over the past decades, most research was done on thin-film type devices, with performance and repeatability often limited by film quality and thickness. For example, [16] has shown the Q factor of a 4H-SiC cantilever beam is 10X higher than that of a 3C-SiC cantilever beam, by improving the film growth method to achieve fewer stacking faults and dislocations. In this work, we use a SiC-on-Insulator (SiCOI) process to enable fabrication of thick monocrystalline 4H-SiC devices in a similar way to conventional silicon-on-insulator (SOI) MEMS devices. The thick 4H-SiC layer results in a larger mass and a higher transduction efficiency compared to thin film devices, with low defects and stress. ...

2) missing information

a) diameter and size of chip in Fig. 1b. (3mm diameter and 4-5mm square?)

a) thickness of 4H-SiC (The value is shown in Fig. 3e. The value should be mentioned in the text as well as Fig. 1b.)

b) condition type and resistivity of 4H-SiC: it may be n-type low resistivity layer (around 30mΩcm?).

c) thickness and resistivity of poly-Si layer

Thank you for the comments, all missing information are added accordingly:

The resonator has a diameter of 3mm and the die size is 5x5mm²

The thickness of 4H-SiC is 40μm, this information is shown in Fig.1b as well as the text

The 4H-SiC is n-type with resistivity of 0.015~0.028 Ω-cm

The poly-Si is P doped with resistivity of ~0.01Ω-cm and thickness of ~0.5μm

...Figure 1a&b demonstrate the schematic and cross-section of the 4H-SiC disk gyroscope fabricated on the SiCOI platform. 0.5μm thick polycrystalline silicon (Poly-Si) contacting pads were patterned on the electrode for ohmic contact formation to the 40μm thick wide-bandgap SiC device layer...

... SiCOI wafers are not commercially available, and are fabricated using the following process flow: a bulk 500μm thick n-type SiC wafer, with resistivity between 0.015~0.028 Ω-cm, purchased from Wolfspeed Inc., was first deposited with 3.5 μm TEOS oxide...

... To fabricate the devices on the SiCOI wafer, first, a thin layer of Phosphorus doped polycrystalline silicon with sheet resistivity around 0.01 Ω-cm, was deposited and patterned to define the contacting electrode pads...

3) In Fig. 5g, 0.001/h^{0.5} (a green star) is just expectation. It looks experimentally demonstrated. "Projected" should be added.

Thanks for the suggestion, the Figure is modified accordingly to avoid confusion.

Reviewer #3 (Remarks to the Author):

The author has provided a comprehensive analysis of 4H silicon carbide bulk acoustic wave MEMS gyroscope with ultra-high Q, highlighting its significance in high-performance MEMS gyroscope design. However, there are a few areas that require further clarification.

On the 2nd page, the author mentions that substrate coupling compromises mode degeneracy ("More importantly, the substrate coupling compromises mode degeneracy."). It would be helpful if the author could provide a more detailed explanation of this phenomenon.

Thank you for the comment, the substrate coupling can compromise mode degeneracy, mostly from the process imperfection in the handle layer and support geometry. Also, the mounting condition, PCB stress, or wire-bonding caused stress can also leak into the resonator. Consider if a resonator is "clamped" to the substrate, its frequency and mode shape can be different than a resonator that is "free-floating"

More importantly, the substrate coupling compromises the mode degeneracy. **Process variations like anchoring misalignment and anchor geometry imperfection can introduce excessive stiffness and damping cross-coupling between the two m=3 BAW modes; in addition, mounting conditions and PCB stress could also affect the resonator through such substrate coupling.** Contrary to the cubic (100) Si, ...

On the 3rd page, the author states that the achievable Q factor in a capacitive SiC disk resonator is mainly limited by surface roughness on the side wall ("Additionally, previous work has shown that the achievable Q factor in a capacitive SiC disk resonator of similar dimension, with or without a decoupling network, is mainly limited by surface roughhouses on the side wall..."). It would be beneficial if the author could summarize the specific etching techniques employed in this study to improve the smoothness of the SiC disk's side wall and subsequently enhance the Q factor.

Thank you for the comment, there are no "special techniques" used to improve the etching profile rather than tuning the etching recipe. And in our experience the side wall roughness is mostly dependent on the profile and growth rate of the nickel fluoride passivation. We want to reach a sweet point where this passivation grows fast enough to cover the sidewall but not too fast to close the gap. We added a note in the supplementary document of the key parameters for achieving a pristine etching profile:

Supplementary Note 2:

The key to achieving a pristine etching profile is to control the nickel fluoride passivation by-product from the SF₆ and nickel mask reaction. Such passivation profile is mostly dependent on electrode temperature and bias power. A lower temperature and higher bias power typically increase the passivation growth rate, which is helpful in preserving the sidewall smoothness but could also pinch the trench as passivation closes the gap; on the contrary, if the passivation growth rate is too slow, significant damage can be seen on the sidewall as it is directly exposed to the plasma.

In the caption of Figure 4, it is mentioned that a positive TCF was measured from 0~70°C, and plateaued

above 80°C. However, it appears that there might be a typographical error, as it should likely be "a positive TQF" instead of "a positive TCF".

Thank you for catching the error, it has been corrected in the manuscript:

... h, a positive TCQ was measured from 0~70°C ...

Figure 4(d) shows a softening effect and a decrease in resonant frequency with increasing actuation amplitude. It would be helpful if the author could provide an explanation for this unexpected behavior.

Thank you for the suggestion; we also conducted a forward-and-backward sweeping characterization recommended by another reviewer. With backward sweeping, the duffing effect becomes more obvious. Beginning at 5dBm, the output from the backward sweep went higher than the forward sweep, followed by a sudden drop in the output. Such duffing and frequency bifurcation is more pronounced as we further increase the drive power to 10,15, and 20dBm.

This nonlinearity is caused by the large driving amplitude in the parallel plate transducer. For example, at 20dBm, the corresponding driving amplitude is over 1 μ m, which is more than 28% of the 3.5 μ m gap width. A typical capacitive transduced resonator often limits the driving amplitude to less than 5% of its gap size to keep linear transduction.

... To characterize the resonator linearity, which feeds into how hard the resonator can be driven to increase the Coriolis coupling, forward-and-backward frequency sweep characterization was conducted with different driving power using a vector network analyzer (VNA), the results of which are shown in Fig. 4j. Spring softening non-linearity was observed in the resonator starting with an input power of 5dBm, where the backward sweep reached a higher output than the forward sweep, followed by a sudden drop in magnitude at lower frequency. As power increases further, the resonator shows a more pronounced Duffing effect, with an increased frequency bifurcation between forward and backward sweep. Such non-linearity is a result of the large displacement amplitude in a parallel plate capacitive transducer. In gyroscope operation, the largest driving voltage used to actuate the resonator is kept at 0dBm or 650mV peak-to-peak, resulting in a disk vibration amplitude of 160nm as obtained from FEA simulation, which corresponds to 5% of the gap size. Stronger actuation of the resonator will push the vibration amplitude to exceed the linear range in a parallel plate capacitive transducer, not only causing instability in the frequency spectrum but also adding noise to the gyroscope output through high-order flicker noise up-conversion. ...

On the 5th page, the author states, "Considering that the inverse of ARW is linearly proportional to Q and noting that the MNE Ω and ENE Ω scale differently with respect to change in Q, we can conclude that the dominating noise is the electronic noise induced ENE Ω , in agreement with our analytical

derivations". It would be beneficial if the author could elaborate further on this point and provide a clearer explanation of their analytical derivations.

Thank you for the comment, because the measured ARW is inversely proportional to the linear change of Q. While MNEΩ is a function of 1/√Q but ENEΩ is a function of 1/Q, and ARW is a combination of both ENEΩ and MNEΩ, we can therefore conclude the ENEΩ is mostly contributing to the overall ARW

Also to highlight the inverse of ARW is linearly proportional to the Q, we replaced the original Fig. 5e (ARW vs. Q) with 1/ARW vs. Q:

...

$$MNE\Omega = \frac{1}{2\lambda|q|} \sqrt{\frac{4k_B T}{\omega_0 m Q}} \quad (4)$$

$$ENE\Omega = \frac{i_{noise}}{2\lambda Q|q|\epsilon AV_p} g^2 \quad (5)$$

...

The measured scale factor and inverse of ARW with different Q were summarized in Fig.5e and can be used to analyze the dominating noise source in this 4H-SiC BAW disk gyroscope. From Eq. (4) and Eq. (5), Brownian noise induced MNEΩ is inversely proportional to √Q while the inverse of ENEΩ is linearly proportional to Q, considering a linear fit best described the inverse of ARW vs Q change, we can conclude that the electronic noise is the main contributor, in agreement with our analytical derivations in Fig.5d. Therefore, in order to break through the navigation requirement, the primary task is to reduce the input-referred current noise i_{noise} of interface electronics. ...

REVIEWERS' COMMENTS:

Reviewer #1 (Remarks to the Author):

The authors have addressed all the comments and I recommend its publication.

Reviewer #2 (Remarks to the Author):

The authors report high-performance on-chip gyroscope devices using bulk 4H-SiC. They theoretically and quantitatively describe the anticipated benefits of utilizing the superior properties of 4H-SiC. Following this, they fabricate actual gyroscope devices, assessing its resonant characteristics and performance. They successfully demonstrate the advantages of 4H-SiC gyroscopes over traditional Si gyroscopes. The paper opens a new field (gyroscope) of 4H-SiC-based MEMS devices, which gives strong impact to the research communities (4H-SiC MEMS related, gyroscope devices and systems).

The authors revised their paper nicely according to all my questions and suggestions (as well as other reviewers' comments). The paper was greatly improved. The revised paper is suitable for publication.

Reviewer #3 (Remarks to the Author):

I have no further questions.